# Comparative Morphology of the Digestive Tract of African Bush Fish (*Ctenopoma acutirostre*) and Paradise Fish (*Macropodus opercularis*) Inhabiting Asian and African Freshwaters

**DOI:** 10.3390/ani13162613

**Published:** 2023-08-13

**Authors:** Dobrochna Adamek-Urbańska, Maciej Kamaszewski, Wiktoria Wiechetek, Rafał Wild, Julia Boczek, Adrian Szczepański, Jerzy Śliwiński

**Affiliations:** Department of Ichthyology and Biotechnology in Aquaculture, Institute of Animal Science, Warsaw University of Life Sciences, Ciszewskiego 8, 02-786 Warsaw, Poland

**Keywords:** labyrinth fish, digestive tract, fish histology, environmental plasticity

## Abstract

**Simple Summary:**

The diets of ornamental aquaculture and aquarium fish usually differ significantly from the diets and types of diets present in natural habitats. These differences can significantly affect the health and welfare of the reared animals. One popular group of aquarium fish is the Anabantoidei fish, which can breathe atmospheric air using an additional respiratory organ—the labyrinth. These fish are commonly bred for both ornamental and consumption purposes. The structure of the digestive system of labyrinth fish such as African bush fish (*Ctenopoma acutirostre*) and paradise fish (*Macropodus opercularis*) is currently not known. In the present experiment, the fishes were studied and characterized based on their morphological and histological body structures. The results obtained suggest that although both species are generally regarded as omnivorous, their digestive tract structures have some differences, including the structure of the mouth, mouth cavity and esophagus, which indicate that the African bush fish is much better adapted to carnivorous feeding than the paradise fish. The results obtained will enable better feeding strategies to be developed for these Anabantoidei, thus improving their health in both commercial and domestic breeding.

**Abstract:**

Anabantidae is a large and diverse group of fish cultured both under aquaculture conditions and as a hobby. These fish share a common structural feature in the form of an additional respiratory organ. Despite the enormous availability of these fish worldwide, little is known about their feeding preferences in husbandry and their influence on homeostasis under both industrial and domestic conditions. This study describes, for the first time, the structure of the digestive tracts of two Anabantoidei fishes: African bush fish (*Ctenopoma acutirostre*) and paradise fish (*Macropodus opercularis*). The overall structure of the digestive tract and its histological structure were analyzed and compared in both fish species. Physiological predispositions indicated a predominance of omnivorous fish traits in *M. opercularis* in contrast to *C. acutirostre,* which has several morphological traits indicating greater adaptation to carnivory, particularly ichthyophagy. The results obtained will allow further research to be conducted in the future to optimize the nutrition and feeding of these fish and to develop appropriate dietary recommendations.

## 1. Introduction

Labyrinths (Anabantoidei) are the most famous group of fish in the world, particularly guramis (Osphronemidae) and Siamense fighting fish (*Betta splendens*), which are famous for their aesthetic qualities [1]. Easy to breed at home, these fish are often chosen not only by hobbyists of labyrinthine fish but also by beginners in the aquarium hobby. The main reason is their intense coloration, small body size, and interesting breeding behavior, unlike other fish groups. Ornamental fish aquaculture is gaining an increasing number of followers every year owing to its high profitability compared with consumer fish aquaculture. It is estimated that the market value of this animal production sector is $15–20 trillion/per annum and will continue to grow rapidly in the coming years [2,3]. This branch of aquaculture includes not only the reproduction of aquarium fish but also those that are caught in the wild. Aquarium fish are of great interest to amateurs, and in some cases, to scientists as well. Excellent examples include zebrafish (*Danio rerio*), medaka (*Oryzias latipes*), and turquoise killifish (*Nothobranchius furzeri*), which are the subjects of molecular, physiological, and neurodegenerative research (e.g., [4,5,6,7]). The suitability of aquarium fish includes not only their use in the above-mentioned categories of research but also their use in nutritional studies whose results may have later application on commercial fish.

The suborder Anabantoidei includes the following families: Anabantidae (about 30 species), Helostomatidae (1 species), Osphronemidae (120 species) with subfamilies Osphroneminae, Luxiocephalinae, and Macropodusinae [8,9]. There is only one species in the Helostomatidae family—*Helostoma temmincki*. All fish from this suborder are characterized by the presence of a labyrinth, which is an auxiliary respiratory organ transformed from gill arches [10]. Fish from the family Anabantidae are found both in Africa and Asia, while the others are found only in Asia, where they occupy various ecological niches. A common feature of the environment in which they occur is the low oxygen content in the water [1,11,12,13,14,15]. 

Ctenopoma is native to Africa Anabantoidei, the bush warbler from the Congo region of central Africa [16]. These fish are animals that swim close to the bottom slowly and are more active at night than during the day. They are not only amateur-bred but are also produced for further distribution around the world. Much better production results are achieved when fed with natural food than with commercial feed; however, the diet does not significantly affect the survival of juvenile stages [15,16]. As predatory fish, Ctenopoma is more likely to eat natural food than fodder, regardless of its composition. These fish are mainly insectivores; however, they can pray even on small vertebrates [17,18,19].

Macropodusinae are small territorial fishes, bred both as an ornamental species and for behavioral research. Their dynamic character (agonistic), similar to betta fish, is used in southeast Asia to conduct fish fights. They come from Taiwan, southern China, northern Vietnam, and the Ryukyu Islands [12]. Their production is much simpler than that of Ctenopoma due to the rapid achievement of sexual maturity and sexual dimorphism, which facilitates breeding. However, the rearing of juveniles, as with all Anatantoidei fish, is challenging due to the requirements for feeding the larvae with natural food, without which their growth rates and survival rates drastically decrease [20]. The best results in rearing juvenile Anabantoidei have been reported with live food [20,21,22,23], which is, unfortunately, more difficult to obtain and use in production than commercial food. Feeding aquarium fish, especially their larval and juvenile stages, requires further studies on the physiology of the gastrointestinal tract and the optimization of not only the composition of the feed itself but also the feeding strategy.

The structure and organization of the digestive tract in fish differ depending on the type of food ingested by these organisms. In predatory carnivorous fish such as *Ctenopoma acutirostre* or the paradise fish (*Macropodus opercularis*), a significant difference in the length of the digestive tract can be observed compared with herbivorous fish such as the kissing gourami (*Helostoma temminckii*). The digestive tracts of plant- or plankton-eating fish are much longer than those of carnivorous and omnivorous fishes [24]. Differences in the structure of the digestive tract of carnivorous and herbivorous fish include not only the variation in the total length of the digestive tract but also its histological structure, in particular the structure of the mouth and esophagus, and the expression of secretory cells, including neuroendocrine cells.

Due to the growing interest in fishkeeping, both in their countries of origin and for export, it is important to develop optimal conditions for rearing and breeding labyrinths. Ensuring optimal production and breeding conditions will not only benefit animal welfare but will also have a positive impact on their economic viability. The development of appropriate and welfare-friendly methods and strategies for feeding juveniles and adults requires basic data on the structure of the digestive tract. In the fish families Anabantidae and Osphronemidae, two species of the most commonly farmed fish of great economic importance were selected and their anatomies described and the structures of their digestive tracts compared.

## 2. Materials and Methods

The study material consisted of 15 specimens of *Ctenopoma acutirostre* and *Macropodus opercularis* collected in 2017–2019. Fish were euthanized with MS222 (130 mg/L) and then weighed and measured (Table 1). Full lengths of the alimentary canals of 7 individuals from each species were dissected and measured to evaluate the length of the intestine sections and then fixed in Bouin’s solutions. Whole fish (n = 8, both species) were fixed in Bouin’s fluid after the abdominal incision. The obtained material was subjected to a standard histological procedure: dehydration, clearing, and embedding in paraffin. Samples were sectioned longitudinally to a thickness of 6µm on Leica microtome RM2265 (Leica Biosystems, Nussloch, Germany). The slides were routinely stained using hematoxylin and eosin (HE) stains. To visualize and differentiate mucosal cells, a combined technique was used to stain acidic mucosal cells with Alcian blue (pH = 2.5) in combination with the staining of neutral mucosal cells with periodic acid and Schiff’s reagent. Microscopic analysis was conducted using a Nikon Eclipse NI-E microscope with a Nikon DS-Fi3 camera and NIS Elements AR software (Nikon, Tokyo, Japan). Statistical differences in IL/SL were calculated in Statistica (Statsoft, Tulsa, OK, USA) using a t-student test.

## 3. Results

The general structure of the digestive tract was similar in both fish species. A large superior, protrusible mouth was found in *C. acutirostre*. The following sections of the gastrointestinal tract were specified: headgut from the superior mouth to the esophagus, stomach, intestine from the pyloric caeca opening, sectioned to the anterior, mid-, and posterior parts (Figure 1). General morphometric comparisons showed that the IL/SL index was significantly higher in *M. opercularis* than in *C. acutirostre*. The gut length calculated based on standard length was also higher in this species (Table 2).

### Headgut

The opening of the digestive tract begins with the mouth covered by nonkeratinized squamous epithelia, with teeth localized on the lips (Figure 2). These teeth have long roots arising from the maxillary and mandible bones. In *C. acutirostre*, the jaw teeth are much more developed and prominent than in *M. opercularis* (Figure 2A,B). The mouth is lined with thin multi-layered epithelia with numerous acidic mucous cells and singular taste buds (Figure 2C,D). Oral valves are present behind the lips of both species.

In *C. acutirostre*, the well-differentiated tongue was characterized by mucous cells and individual taste buds. The latter were found singly throughout the oral cavity, including the interior of the gills and esophagus. In *M. opercularis*, the tongue was merely a protuberance in the oral mucosa, with cartilaginous support similar to that of *C. acutirostre* (Figure 3). The tongue core in both species was composed of hyaline cartilage and a partial PAS-positive intercellular matrix. Mucosal cells and taste buds lined the mouth cavity and esophagus, as in *C. acutirostre* but were neutral, mixed, and acidic, unlike in *M. opercularis*. 

The esophagus also contains pharyngeal teeth between the strongly corrugated mucosal divides. The teeth grow from a single dental bone plate. In the subsequent sections, the secretory characteristics of the cells changed to mixed mucous cells (acidic and neutral) and then exclusively to neutral in the stomach (Figure 4).

The stomachs of both *C. acutirostre* and *M. opercularis* resemble sacs similar to those of monogastric terrestrial vertebrates. The epithelial cells are covered by neutral mucins-stained magenta in AB/PAS (Figure 4A,B). The stomach of *M. opercularis* has less folding and is more complex than that of *C. acutirostre*. Two short pyloric caeca were observed behind the gastric openings of both species, with those in *M. opercularis* being larger than those in *C. acutirostre* (Table 2). Similarly, to follow sections of the digestive tract the epithelium cells were higher compared to the stomach with singular mucosal cells produced acidic and mixed (neutral and acidic) mucins (Figure 4C–J). Singular yellow–brownish macrophages were present in some intestine folds. The pH of mucosal cells in *C. acutirostre* and *M. opercularis* differed. In *C. acutirostre*, the pH of mucosal cells was mostly highly acidic or mixed. In *M. opercularis*, the pH of mucosal cells was also acidic but with more frequently mixed or neutral pH, in particular in the anterior intestine.

## 4. Discussion

Ornamental fish kept at home or in aquaculture, for example, the well-known commercial fish species, require optimal husbandry conditions. Optimal husbandry conditions demand knowledge and a basic understanding of the anatomy, morphology, and physiology of the animals. For fish and aquatic animals with extremely varied food-acquisition strategies, knowledge of the structure of the digestive tract is particularly important and allows subsequent research into the physiology of this system. Such knowledge can help avoid feeding and welfare problems, which often occur during fish breeding in home aquaria [3,25].

The studied fish species had similar gastrointestinal structures, though there were significant differences between them. The overall structures of the digestive tracts of *C. acutirostre* and *M. opercularis* indicated differences in both the ability to obtain food and to swallow and subsequently digest it. Gut length, which contributes to absorption efficiency and is correlated with total fish length, is strongly correlated with diet [26,27,28,29]. The digestive tract is shorter in carnivorous fish than in omnivorous or herbivorous fish [28,30]; this was confirmed in this study. The comparative analysis conducted in the present study showed that the digestive tract of *C. acutirostre* is shorter than that of *M. opercularis.* These observations indicate a greater adaptation to a carnivorous diet compared to the omnivorous *M. opercularis*. Previous studies on the feeding of *C. acutirostre* under aquaculture conditions showed that the best growth rates were achieved with live natural food (silkworms) [15,16]. However, it is important to take into account the significant variations in food preferences depending on the location of *C. acutirostre* due to the diverse ecological niches occupied by this species. As an example of such variation, another species, *Ctenopoma pathereri* Gunther, known—similarly to *C. acutirostre*—as an omnivore, has greater herbivorous tendencies in its habitat in the Oluwa River, Ondo State, Nigeria [31].

The morphological structure of the digestive tract, starting with the superior mouth opening, also indicated differences in diet [32]. The mouth of *C. acutirostre* was large and deeply indented, indicative of the strong structure and considerable length of the maxillary, premaxillary, and dentary bones. This construction pattern also promotes the protrusion of the snout when grasping prey. In contrast to *C. acutirostre*, the mouth of *M. opercularis* is smaller, even delicate, and does not show a deep indentation. In fish living in waters with less light and poor visibility, as is periodically the case for Anabantoidei, taste buds located on the mouth facilitate food searching and recognition [33]. Their role in low-transparency environments is extremely valuable in enabling not only the identification of food but also the assessment of its tastiness. In predatory fish, the mouth is more extensive and is equipped with teeth that help to capture and hold prey, and further assist with sucking in water along with the prey. These teeth can have highly variable shapes that are closely linked to their diets. In carnivorous fish, the teeth are narrow, long, and sharp; in omnivorous or predominantly herbivorous fish, they are much lower, broad, and blunt. The localized teeth in the two species studied were found both on the lips and in the esophagus localized on the bony plate. Given the mobility and width of the mouth opening of *C. acutirostre,* the teeth and the central valve help to catch, suck in, and impede the retraction of food. The design of the mouth and mouth opening of omnivorous *M. opercularis* and *C. acutirostre* indicate that the mouth opening of *C. acutirostre* is significantly more adapted to capturing larger prey. Since *C. acutirostre* in the wild prey not only on insects—similar to *M. opercularis*—but also on arachnids and even frogs and other fish, the occurrence of teeth in the lips indicates their adaptation to hunting live food [34,35].

In some fish species [36,37], a tongue is present in the mouth cavity, but its structure and mobility are significantly reduced, similar to terrestrial vertebrates. The tongues of Anabantoidei fishes are characterized by less mobile and complex structures compared with the tongues of terrestrial vertebrates (lack complex undulations and differentiated taste buds). Despite this, this organ has functions associated not only with swallowing and swiping food but also identification of taste stimuli through taste buds located on its surface [37]. Thus far, the fish’s tongue has been described in the literature as a thickening at the bottom of the mouth [33,37,38,39]; however, in recent years, increasing data in the literature indicates the importance of this organ in food intake [40], and its identification and recognition of taste due to the numerous taste buds present on it [37,38]. The tongues of the Anabantoidei examined here were characterized by a convergent overall histological structure; however, the tongue of *C. acutirostre* was much longer and more freely attached to the floor of the oral cavity, leading to the conclusion that it is probably more mobile compared with the deeply embedded, small tongue of *M. opercularis*. Due to its apparent lack of muscular tissue, the motility of the tongue of *C. acutirostre* does not consist of conscious and deliberate movement as is the case in e.g., mammals; instead, it uses it—together with the oral valve—to swallow, restrain, and further force through the swallowed food, similar to other ichthyophages. The rigidity of this organ is due to the structure of its core, which is made up of supporting connective tissue. The staining of the tongue core using HE and AB/PAS indicated that it is an osteocartilaginous skeleton, similar to that observed in another ichthyophage, the northern pike (*Esox Lucius*) [41]. In contrast, none of the specimens examined revealed the presence of tongue surface undulations with teeth growing from the tongue surface that Levanti et al. described in sea bass (*Dicentrarchus labrax*), seabream (*Sparus aurata*), and white seabream (*Diplodus sargus sargus*) [37].

The food passed through the pharyngeal teeth and entered the stomach. In many fish species, the stomach may not be present at all—its functions are performed by the morphologically transformed anterior intestine. However, in many fish species where the stomach is absent, partial compensation for this organ is also provided by expanded wide pharyngeal or gizzard teeth, which grind the food more thoroughly than in other fish species, facilitating further digestion by the transformed initial intestinal region [42,43]. In the case of Anabantoidei representatives, the presence of a stomach is confirmed in the present study, as well in studies of dwarf gourami (*Colisa lalia*) [44], giant gourami (*Osphonemus goramy* Lacepede, 1801) [45], and snakeskin gourami (*Trichopodus pectoralis*) [46]. In carnivorous fish, the stomach mucosa is more strongly developed to cope with the digestion of more complex food containing large amounts of proteins. Considering this aspect, the stomach of *C. acutirostre* was more strongly developed compared with that of *M. opercularis*. Nonetheless, pyloric caeca were present in both studied species, with their stomachs secreting the enzymes necessary to partially digest the nutritional matter.

The pyloric appendages are an evolutionary solution to increase the absorptive surface area of the digestive tract. Their structure is very similar to that of the distal intestinal tract. Their number is a species-specific trait, varying from a few appendages in fish such as spotted snakehead (*Channa punctata*) to several hundred in Atlantic cod (*Gadus morhua*) [26]. They are absent in stomachless fish [43,47,48]. Previous studies have not shown a correlation between the number of pyloric caeca and the type of food [26]; however, their numbers are lower in omnivorous and herbivorous species than in carnivorous species [49] due to the need to digest chemically and structurally more demanding foods. The presence of only two pyloric caeca suggests that the efficiency of digestion in the earlier sections of the digestive tract is probably sufficient to ensure proper absorption of some nutrients in the first section of the intestines. No significant differences were observed in the structure of the further sections of the digestive tract, and the normal layered structure of the intestines consisting of mucosa, submucosa, and musculature—characteristic of all vertebrates—was found. The only difference was stronger AB-positive staining of the mucosal cells of the anterior and middle intestines of *C. acutirostre*, indicating the more acidic characteristic of the produced mucus. The mucous cells in the gastrointestinal tract perform many functions; the key functions are facilitating the passage of food, protecting the epithelium from mechanical damage, actively participating in the fight against microorganisms entering the gastrointestinal tract with food, and creating a more favorable environment for nutrient absorption.

## 5. Conclusions

The present study identified differences in the structures of the digestive tract of representatives of *Anabantoidei* fishes belonging to two different families that are considered omnivorous. *C*. *acutirostre* was characterized by improved adaptation to obtaining and processing carnivorous foods. The absence of differences in the histological structure of the intestine does not imply the absence of differences in their physiology. The results of this study provide basic knowledge of the morphology and histology of the gastrointestinal tract of these fish, providing a solid basis for further research into the physiology of the gastrointestinal tract and optimization of feeding in both species studied. However, further research is needed to demonstrate physiological differences in the two species that would allow the development of more optimal diets that could probably benefit survival rates, particularly in larval and juvenile stages of the fish where mortality is highest.

## Figures and Tables

**Figure 1 animals-13-02613-f001:**
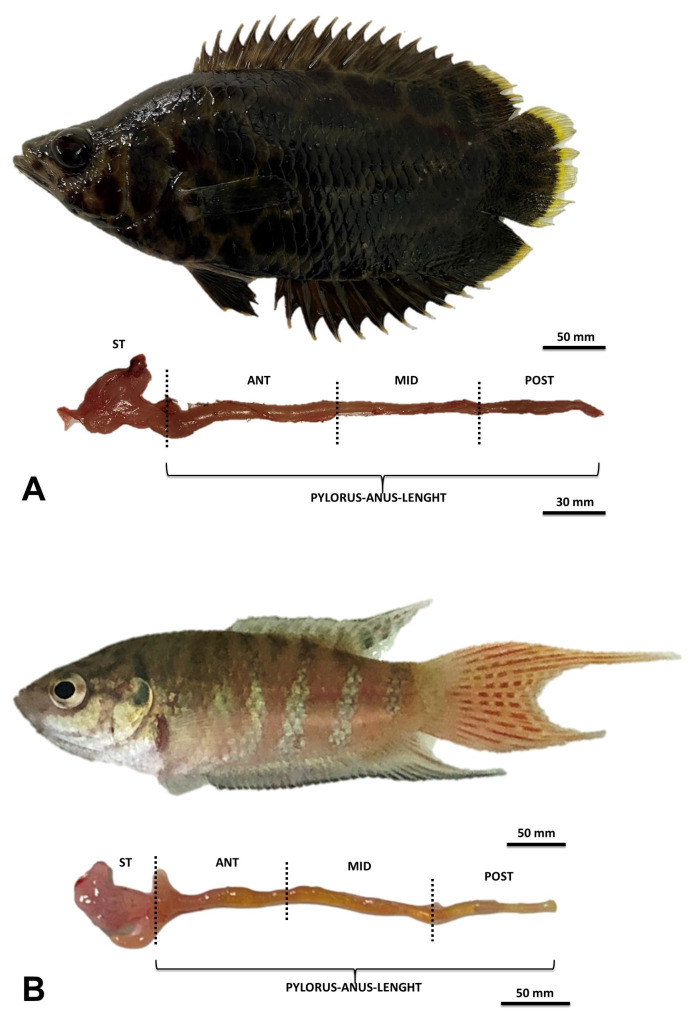
Macroscopic view of the digestive tract of *C. acutirostre* (**A**) and *M. opercularis* (**B**) ST: stomach; ANT: anterior intestine; MID: mid-intestine; POST: posterior intestine.

**Figure 2 animals-13-02613-f002:**
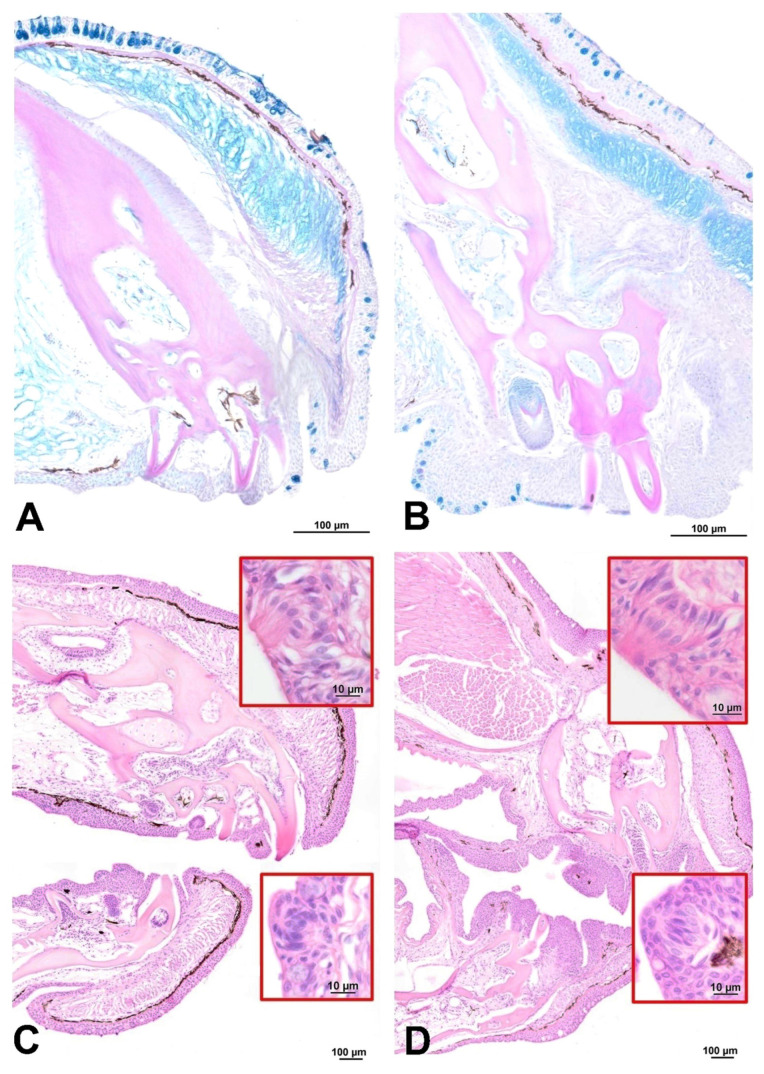
Transverse sections of the mouth in (**A**,**C**) the African bush fish and (**B**,**D**) the paradise fish. In cross-sections through the lower lips, the weakly staining PAS-positive bone from which the teeth emerge is evident. In African bush fish (**A**), this is a uniform block of tissue, unlike in the paradise fish (**B**). Taste buds located on both the external and internal parts of the mouth cavity were also visible based on HE staining (magnifications in red frame); (**A**,**B**) AB/PAS staining; (**C**,**D**) HE staining.

**Figure 3 animals-13-02613-f003:**
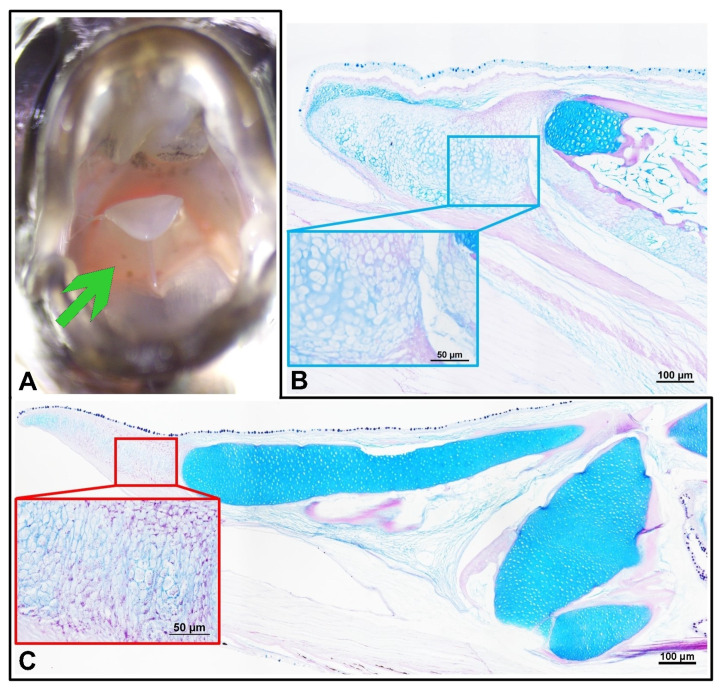
In African bush fish ((**A**,**C**) in the black frame), the tongue structure was visible macroscopically (green arrow), with the oral valve in the upper part of the mouth. The histological structure of the tongue showed that the whole structure is based on two fragments of hyaline cartilage (**B**) with the front growing cartilage full of magenta-positive droplets (magnification in the red frame). The tongue of the paradise fish is structured with hyaline cartilage and singular areas of ossification (magnification in the blue frame).

**Figure 4 animals-13-02613-f004:**
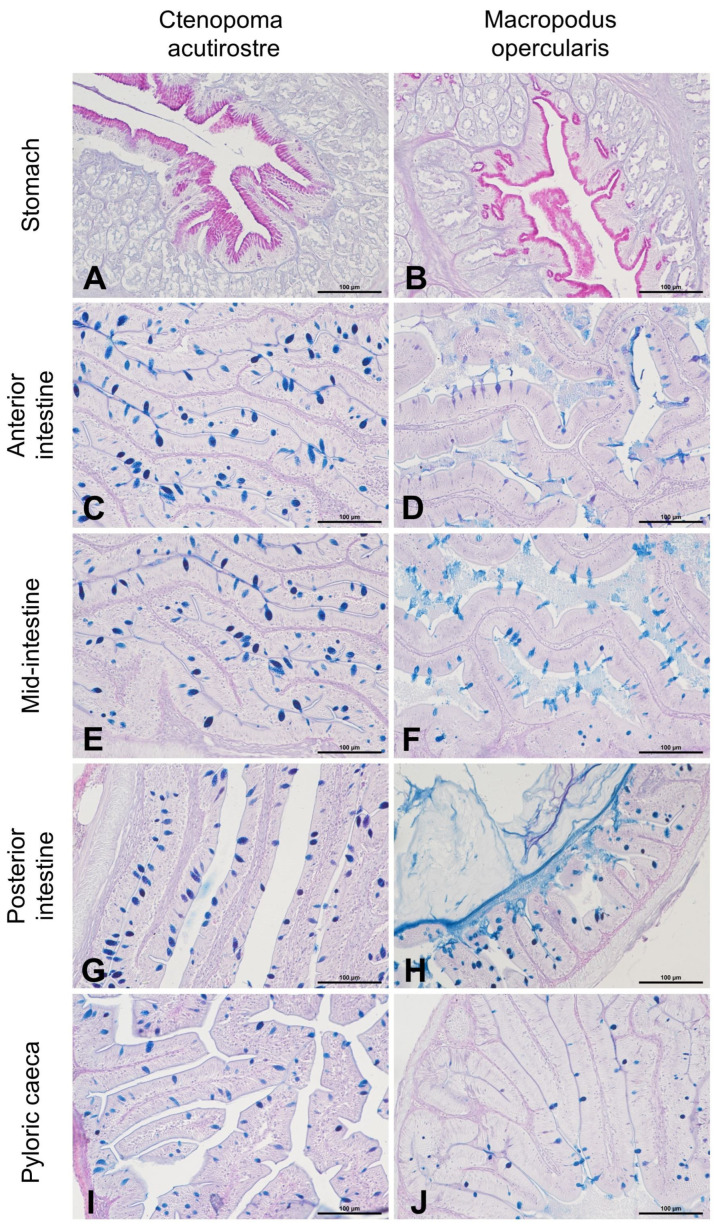
Histological structure of digestive tract sections: stomach (**A**,**B**), anterior (**C**,**D**), mid-intestine (**E**,**F**), posterior intestine (**G**,**H**), and pyloric caeca (**I**,**J**). The neutral mucins in the stomach mucosa are stained magenta, which is visible in the apical part of the cells. In the following sections, mucosal cells are stained blue; AB/PAS staining.

**Table 1 animals-13-02613-t001:** Morphometric data for investigated fish (n = 15).

	Total Length (cm)	Standard Length (cm)	Fork Length (cm)	Body Weight (g)
*Ctenopoma acutirostre*	5.39 ± 0.23	4.58 ± 0.24	0.82 ± 0.09	2.88 ± 0.56
*Macropodus opercularis*	5.47 ± 0.49	3.68 ± 0.18	1.79 ± 0.37	1.30 ± 0.49

**Table 2 animals-13-02613-t002:** General morphometrics of *C. acutirostre* and *M. opercularis* (SL: standard length; IL: intestine length).

Length (cm)	*Ctenopoma acutirostre*	*Macropodus opercularis*
SL (cm)	4.58 ± 0.24	3.68 ± 0.18
IL (cm)	3.38 ± 0.47	4.14 ± 0.70
Pyloric caeca length (cm)	0.25 ± 0.13	0.41 ± 0.09
IL/SL	0.74 * ± 0.09	1.13 * ± 0.14
Range of IL/SL	0.59–0.83	0.93–1.31
% of body length	59–83	93–131

The data represent mean ± SD of n = 7 fish of both species. * statistical differences (*p* < 0.001) between species.

## Data Availability

Data which was not presented in this manuscript or supplementary material is available on request from the corresponding author.

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
