# Peer review of "Comparative Morphology of the Digestive Tract of African Bush Fish (Ctenopoma acutirostre) and Paradise Fish (Macropodus opercularis) Inhabiting Asian and African Freshwaters"

_animals, 2023, doi:10.3390/ani13162613_

Round 1
Reviewer 1 Report
Dear Authors
The authors examined the comparative morphology of the digestive tract of African bushfish (Ctenopoma acutirostre) and paradise fish (Macropodus opercularis). The authors histologically examined the digestive tract of two fish and compared them with other fish. The results of the paper are potentially interesting for Animals and it is well-aimed. However, there seem to be areas of scientific problems in the manuscript. This study may be a preliminary study for two fish species, but it cannot be published as it is. Before decision on the manuscript, the author should give an appropriate answer to the major comments at below.
MAJOR REMARKS
1. English of the manuscript should be revised by a native English speaker or editing services.
2. Page 2, line 79-80. The authors say that the alimentary canal length of herbivorous fish is shorter than that of omnivorous or carnivorous fish. This sentence is quite wrong. Please correct.
3. Alcian Blue stains different glycoconjugates at different pHs. Therefore, what is the pH value of the alcian blue stain?
4- The authors compared some values with each other. Which statistical method did you use?
5- How did you specify the parts of the intestine as anterior, mid, and posterior intestine?
MINOR COMMENTS
- please add the number of samples (n) used in Table 1.
- Page 3, lines 92-93. The values are close to each other. Which fish did you fix whole?
-page 3, line 95. Please correct as 6µm.
-Page 3 line 96. Please correct as Periodic Acid Schiff.
-Table 1. Sometimes commas and sometimes dots are used in numerical values.
Please see the journal author guide.
Page 8, line 208. Correct. Localised
The spelling of some words in the manuscript is incorrect.
In the introduction and discussion, the meanings of some sentences are used opposite to each other.
Author Response
The authors examined the comparative morphology of the digestive tract of African bushfish (Ctenopoma acutirostre) and paradise fish (Macropodus opercularis). The authors histologically examined the digestive tract of two fish and compared them with other fish. The results of the paper are potentially interesting for Animals and it is well-aimed. However, there seem to be areas of scientific problems in the manuscript. This study may be a preliminary study for two fish species, but it cannot be published as it is. Before decision on the manuscript, the author should give an appropriate answer to the major comments at below.
On behalf of all of the authors, I would like to thank the reviewer for helpful comments and constructive remarks. We took into consideration all of the modifications which was suggested and have revised the manuscript accordingly. Our responses to the reviewer’s comments and suggestions are provided below.
MAJOR REMARKS
- English of the manuscript should be revised by a native English speaker or editing services.
Manuscript was revised by Native Speaker.
- Page 2, line 79-80. The authors say that the alimentary canal length of herbivorous fish is shorter than that of omnivorous or carnivorous fish. This sentence is quite wrong. Please correct.
The sentence was corrected.
- Alcian Blue stains different glycoconjugates at different pHs. Therefore, what is the pH value of the alcian blue stain?
The pH of alcian blue was added to material and methods.
4- The authors compared some values with each other. Which statistical method did you use?
Thank you for highlighting this issue. Among the studied parameters, only the IL/SL index was compared in value range. Due to the specificity of the parameter, a one-way ANOVA with Tukey's post hoc test for an equal number of samples was used. Statistically significant differences are highlighted in Table 2, P value information is provided under the table and a notation of the analysis used is included in the methodology.
5- How did you specify the parts of the intestine as anterior, mid, and posterior intestine?
The intestines sections were specified for parts, according to literature data, as well as general knowledge and further microscopic analyses, which allowed to determine distinctive cytological features of intestinal duct as we presented in Figure 1 and 4. However, several possibilities for the subdivision of the fish intestines can be found in the literature. Some authors state that the anterior part is already the pyloric caeca, followed by the mid and the posterior intestine (Ray and Ringø, 2014; Kamalan et al., 2020). Other authors divide the digestive tract as in this paper (Ostaszewska et al., 2008; Kamaszewski & Ostaszewska, 2014; Alabssawy et al., 2019; Goodrich et al., 2020). A third possibility is to divide only into anterior and posterior intestines in fish, in which the histological and morphological structure of the midgut does not differ from the anterior intestine (Vieira-Lopes et al., 2013; Egerton et al., 2018). Nevertheless, in the present study, due to the presence of both a stomach and pyloric appendages, and differ structure of mid intestine compared to anterior, we used the most common division into anterior mid- and posterior intestine.
References
- Alabssawy, A. N., Khalaf-Allah, H. M., & Gafar, A. A. (2019). Anatomical and histological adaptations of digestive tract in relation to food and feeding habits of lizardfish, Synodus variegatus (Lacepède, 1803). The Egyptian Journal of Aquatic Research, 45(2), 159-165.
- Egerton, S., Culloty, S., Whooley, J., Stanton, C., & Ross, R. P. (2018). The gut microbiota of marine fish. Frontiers in microbiology, 9, 873.
- Goodrich, H. R., Bayley, M., Birgersson, L., Davison, W. G., Johannsson, O. E., Kim, A. B., ... & Wood, C. M. (2020). Understanding the gastrointestinal physiology and responses to feeding in air‐breathing Anabantiform fishes. Journal of Fish Biology, 96(4), 986-1003, doi.org/10.1111/jfb.14288
- Kamalam, B. S., Rajesh, M., & Kaushik, S. (2020). Nutrition and feeding of rainbow trout (Oncorhynchus mykiss). Fish nutrition and its relevance to human health, 299-332.
- Kamaszewski, M., & Ostaszewska, T. (2014). The effect of feeding on morphological changes in intestine of pike-perch (Sander lucioperca L.). Aquaculture international, 22, 245-258.
- Ostaszewska, T., Dabrowski, K., Hliwa, P., Gomółka, P., & Kwasek, K. (2008). Nutritional regulation of intestine morphology in larval cyprinid fish, silver bream (Vimba vimba). Aquaculture Research, 39(12), 1268-1278.
- Ray, A. K., & Ringø, E. (2014). The gastrointestinal tract of fish. Aquaculture nutrition: Gut health, probiotics and prebiotics, 1-13, doi.org/10.1002/9781118897263.ch1.
- Vieira-Lopes, D. A., Pinheiro, N. L., Sales, A., Ventura, A., Araújo, F. G., Gomes, I. D., & Nascimento, A. A. (2013). Immunohistochemical study of the digestive tract of Oligosarcus hepsetus. World Journal of Gastroenterology: WJG, 19(12), 1919.
MINOR COMMENTS
- please add the number of samples (n) used in Table 1.
The number of samples was put in brackets (n=15)
- Page 3, lines 92-93. The values are close to each other. Which fish did you fix whole?
For comparative analysis of intestinal length, the digestive tracts were taken from 7 individuals of each species. The remaining eight fish of each species were also fixed whole to illustrate the initial section of the digestive tract. The sampling methodology was specified and described in more detail in Materials and methods.
-page 3, line 95. Please correct as 6µm.
The sentence was corrected.
-Page 3 line 96. Please correct as Periodic Acid Schiff.
The sentence was corrected.
-Table 1. Sometimes commas and sometimes dots are used in numerical values.
We apologize for the omission. Table 1 has been corrected, and the following numerical values have been verified for correct transcription.
Please see the journal author guide.
Page 8, line 208. Correct. Localised
Comments on the Quality of English Language
The spelling of some words in the manuscript is incorrect.
In the introduction and discussion, the meanings of some sentences are used opposite to each other.
Manuscript was revised by Native Speaker.

Reviewer 2 Report
The manuscript (animals-2478705) compared the digestive tract features of two Anabantidae fish: African bushfish Ctenopoma acutirostre and paradise fish Macropodus opercularis. The results showed that, although both species were traditionally regarded as omnivorous, their digestive tract features clearly indicate that the African bushfish is much better adapted than the paradise fish for carnivorous feeding. The experimental design made sense, and the manuscript is generally well written. Before publication, the following issues should be addressed:
Major concerns
(1) The information of the sampling site should be detailed. The authors only stated that the specimens were sampled in 2017-2019. The specific site and season (month) should be provided. To better compare the digestive features, I personally suggested that the specimens should be collected at similar regions at the same season.
(2) The word “significantly” was used in the results and discussion section. However, statistics was not run in this study, even for the morphological digital values.
(3) If the authors could provide the natural food composition in the digestive tract of the C. acutirostre and M. opercularis, the conclusion should be more convincing.
Minor issues
Title: Provide the name of the target fish species
L95: “um” should be replaced with “μm”
Table 1: “,” should be replaced with “.”
Figure 1: OS and PC were not indicated in the Figures.
The conclusion was too long, thus should be refined.
Author Response
The manuscript (animals-2478705) compared the digestive tract features of two Anabantidae fish: African bushfish Ctenopoma acutirostre and paradise fish Macropodus opercularis. The results showed that, although both species were traditionally regarded as omnivorous, their digestive tract features clearly indicate that the African bushfish is much better adapted than the paradise fish for carnivorous feeding. The experimental design made sense, and the manuscript is generally well written. Before publication, the following issues should be addressed:
On behalf of all of the authors, I would like to thank the reviewer for his/her helpful comments and constructive remarks. We took into consideration all of the modifications which were suggested and have revised the manuscript accordingly. Our responses to the reviewer’s comments and suggestions are provided below.
Major concerns
(1) The information on the sampling site should be detailed. The authors only stated that the specimens were sampled in 2017-2019. The specific site and season (month) should be provided. To better compare the digestive features, I personally suggested that the specimens should be collected at similar regions in the same season.
The fish species under study are commonly farmed as ornamental fish in Europe. The study was conducted on fish obtained from local markets. The fish were of breeding origin in Europe, they were not caught, therefore we cannot indicate the exact location. Furthermore, there were fish species kept under ornamental aquaculture conditions in temperature-stable and controlled aquaria, the season was not relevant in this case.
(2) The word “significantly” was used in the results and discussion section. However, statistics were not run in this study, even for the morphological digital values.
Thank you for pointing out this error. Indeed, the text gives the impression that we are comparing numerical values; however, this only refers to parameters related to body length and IL/SL. IL/SL was also estimated statistically, as suggested by both reviewers. Therefore, the results and the discussion have been carefully edited to remove confusing or misleading information.
(3) If the authors could provide the natural food composition in the digestive tract of C. acutirostre and M. opercularis, the conclusion should be more convincing.
It would definitely make it much easier to support our conclusions by carrying out such analyses. However, because these were aquarium-kept fish, they were fed commercial foods that would not show significant differences. The aim of our work was to verify whether the morphological structure of the digestive tract of the studied species itself differs in any way, or whether it is the same since both species are commonly recognized as omnivorous. In the near future, we plan to apply for a project in which we could examine the impact of the applied feeding strategy in the F1 generation, in order to evaluate, independently of the parents' diet, the effect of natural, commercial and combined feeds on the structure and physiology of the digestive system, especially the expression of regulatory proteins of the sense of hunger and satiety in these fish.
Minor issues
Title: Provide the name of the target fish species
The title was corrected.
L95: “um” should be replaced with “μm”
The sentence was corrected.
Table 1: “,” should be replaced with “.”
The sentence was corrected.
Figure 1: OS and PC were not indicated in the Figures.
Figure 1 was corrected.
The conclusion was too long, thus should be refined.
We have revised the summary, it has been shortened and compiled

Round 2
Reviewer 1 Report
The authors have made some revisions. However, the statistical problem that I mentioned in the first revision still persists. The researchers used the ANOVA (Analysis of variance) method used in multiple comparisons for the two groups. ANOVA and the post hoc test they use are incorrect for this study. This problem needs to be corrected before the manuscript is accepted.
Author Response
Dear Reviewer,
We apologise most sincerely for our error.
I'm afraid I do not fully understand your comment. The parameter test used in our study is commonly used in this type of research. We used the ANOVA – one – way ANOVA, not two or multiple ANOVA. We chose the Tukey post hoc test because of the need to compare results between less numerous quantitative data. We verified whether we would have obtained different results after using any other parametric test or ANOVA with other post hoc tests. No matter which test we used the results are the same - highly significant differences. Therefore, I am not really aware if you would like us to use another test - if so, which one? I would really appreciate a straightforward hint as I don't appear to understand what you are inquiring about.
Sincerely,
Dobrochna Adamek-Urbańska
Reviewer 2 Report
The authors have well addressed my concerns.
Author Response
Dear Reviewer,
thank you sincerely for your contribution to improving the quality of our work. Your advice was extremely valuable and facilitated our proper revision of the article.
Sincerely,
Dobrochna Adamek-Urbańska
Round 3
Reviewer 1 Report
The article "Comparative morphology of the digestive tract of African bush fish (Ctenopoma acutirostre) and paradise fish (Macropodus opercularis) inhabiting Asian and African freshwaters" has been re-examined. But there still seem to be some problems. In particular, despite the editor's statement, multiple comparison tests are used in the study. Why is the t-test suitable for pairwise comparison not used?
Some minor revisions to the manuscript are shown below.
Although it is a histological study, the method is given very briefly. It could have been written in more detail.
page 3, line 104. please add a dose of MS222.
"individuale" please correct.
page 3, lines 109-110. haematoxylin and eosin (HE), capitalize the letter H.
Write "Nikon DSFi3" as DS-Fi3.
Page 3, lines 122-123. Esophagus written twice, delete one.
The fonts in Table 1 and Table 2 are different, please correct.
The abbreviations and figure descriptions in Figure 1 are inconsistent. The rectum is not shown in the figure, either add or delete it.
Author Response
Dear reviewer,
In the case of methodology, as I mentioned before, we used the one-way ANOVA test due to its best fit for this study. Since this is a debatable matter, we have changed the ANOVA to a student's t-test as you suggested, and the changes have been included in the text of the publication.
We have also made corrections according to your suggestions and developed the description of the research methodology.
We hope that the changes made are satisfactory.
Round 4
Reviewer 1 Report
Dear Authors,
Some corrections have been made in Figure 1, but they are not sufficient. For example, the part shown as ANT in the figure is shown as AI in the description. Likewise, MID is expressed as MI and PI in POST.
In Figure 1, "stomach;" Leave a space after
Author Response
Dear Reviewer,
Thank you for your submitted comments and suggestions. The manuscript has been revised according to them and I hope that in its present form it meets the reviewer's expectations.
Sincerely,
DAU